# Detection of Liver Dysfunction Using a Wearable Electronic Nose System Based on Semiconductor Metal Oxide Sensors

**DOI:** 10.3390/bios12020070

**Published:** 2022-01-26

**Authors:** Andreas Voss, Rico Schroeder, Steffen Schulz, Jens Haueisen, Stefanie Vogler, Paul Horn, Andreas Stallmach, Philipp Reuken

**Affiliations:** 1Institute of Innovative Health Technologies IGHT, Ernst-Abbe-Hochschule Jena, 07745 Jena, Germany; schroederrico@gmx.de (R.S.); steffen.schulz@charite-berlin.de.com (S.S.); 2Institute of Biomedical Engineering and Informatics (BMTI), Technische Universität Ilmenau, 98693 Ilmenau, Germany; jens.haueisen@tu-ilmenau.de; 3UST Umweltsensortechnik GmbH, 99331 Geratal, Germany; 4Clinic for Internal Medicine IV, University Hospital Jena, 07747 Jena, Germany; Gastro@med.uni-jena.de (S.V.); P.Horn@bham.ac.uk (P.H.); andreas.stallmach@med.uni-jena.de (A.S.); philipp.reuken@med.uni-jena.de (P.R.)

**Keywords:** electronic nose, liver dysfunction, cirrhosis, semiconductor metal oxide gas sensor

## Abstract

The purpose of this exploratory study was to determine whether liver dysfunction can be generally classified using a wearable electronic nose based on semiconductor metal oxide (MOx) gas sensors, and whether the extent of this dysfunction can be quantified. MOx gas sensors are attractive because of their simplicity, high sensitivity, low cost, and stability. A total of 30 participants were enrolled, 10 of them being healthy controls, 10 with compensated cirrhosis, and 10 with decompensated cirrhosis. We used three sensor modules with a total of nine different MOx layers to detect reducible, easily oxidizable, and highly oxidizable gases. The complex data analysis in the time and non-linear dynamics domains is based on the extraction of 10 features from the sensor time series of the extracted breathing gas measurement cycles. The sensitivity, specificity, and accuracy for distinguishing compensated and decompensated cirrhosis patients from healthy controls was 1.00. Patients with compensated and decompensated cirrhosis could be separated with a sensitivity of 0.90 (correctly classified decompensated cirrhosis), a specificity of 1.00 (correctly classified compensated cirrhosis), and an accuracy of 0.95. Our wearable, non-invasive system provides a promising tool to detect liver dysfunctions on a functional basis. Therefore, it could provide valuable support in preoperative examinations or for initial diagnosis by the general practitioner, as it provides non-invasive, rapid, and cost-effective analysis results.

## 1. Introduction

Metabolic disorders are sometimes connected with typical odors which can be measured on breath, sweat, or other excreta from humans. Examples are ammonia odor, which is related to renal diseases, and acetone odor, which is related to diabetes.

The beginnings of the use of electronic noses (e-noses) date back to pioneering work by a few research groups, such as Hartman, Wilkens, Dodd, and Moncrieff [1,2,3,4]. Here, the foundation was laid for specific odors to be detectable and, thus, evaluable with suitable electronics and analysis technology. The concept of sampling breath for health monitoring was initially conceived in the 20th century. In 1952, Henderson [5] reported on the increased acetone content of breath samples from young diabetics, promoting an interest in the content of breath [6].

In recent decades, improvements in materials, sensors, electronics, and signal processing technologies have led to a rapid increase in the development and application of e-noses [7,8,9]. E-noses are used, among other things, to analyze, detect, discriminate, classify, and monitor gas components or odors in many fields of science and industry, and are of interest for numerous applications. For example, e-noses are used in the food and beverage industry to monitor processing and determine the quality of the final product [10,11], in pharmaceutical science for formulation development and quality assurance [8], and for air quality monitoring [12]. In addition, e-noses are also used in agriculture, water management, medicine, security systems, and many other fields [13].

In the following, we will only deal with the e-noses that meet Gardner’s definition [14]. He stated that an e-nose is an instrument, which comprises an array of electronic chemical sensors with partial specificity and an appropriate pattern-recognition system, capable of recognizing simple or complex odors. However, unlike other analytical methods, an e-nose does not detect directly specific volatile organic components (VOCs); rather, it builds chemical patterns to form an identity. The sensor array produces output patterns that represent VOCs in the breath (or different substances), and the data processing extracts a set of mathematical descriptors that represent the signature of the breath sample as a pattern [15]. The detection of the input signal occurs depending on the operating principle implemented in the sensor arrays. There are a variety of sensor types used in e-nose technology. These include the following in particular [13,16]:Metal oxide,Conducting polymer,Quartz crystal microbalance,Acoustic wave,Electro-chemical,Catalytic bead,Optical.

Among these available gas sensing methods, semiconducting metal oxide gas (MOx) sensor devices have several unique advantages, such as low cost, small size, easy measurement, durability, ease of fabrication, and low detection limits (low ppm level). In addition, most MOx-based sensors are relatively resistant to poisoning. For these reasons, they have quickly gained popularity and have become the most widely used gas sensors today [13].

E-noses have also been developed for medical applications. Here, e-noses can distinguish between different types of diseases and their severity by analyzing body odor. This includes disease-related metabolic changes especially [17], but any kind of drug consumption [18] can also be detected on the skin surface and/or exhaled breath.

One can show that such e-noses can be successfully used to improve the diagnosis of various diseases, ranging from kidney disease [19] and diabetes [20], to various types of respiratory diseases [21] and carcinoma [22,23], up to heart diseases [24]. These and other studies provide evidence that, after a necessary validation, a cost-effective, portable, and fast working e-nose system could be useful for improved diagnostics and health protection.

The diagnosis of chronic liver disease is usually based on a combination of clinical signs, laboratory parameters, and imaging results [25]. However, this approach has several important weaknesses. First, the prognosis of cirrhosis depends on the structure and function of the liver, but even more important is the occurrence of complications, such as variceal bleeding or infections [26,27]. Second, laboratory values can be influenced by other conditions that present in the same way as cirrhosis, which may lead to misinterpretation. Third, some imaging techniques, such as transient elastography, are influenced by “non-liver” factors, such as central venous pressure [28]. Fourth, there is a great need for an exact measurement of the current patient’s situation to choose the optimal treatment, e.g., if transplantation is needed or a non-hepatic or hepatic surgery must be performed. In current scoring systems, such as the Model for End-stage Liver Disease (MELD) [29], patients with portal hypertension as a decompensating event (ascites, variceal bleeding) are poorly represented due to the nature of the score.

Based on a proof-of-concept study, De Vicentis et al. [30] showed that an e-nose based on piezoelectric gas sensors could be a valid non-invasive instrument for characterizing chronic liver disease and monitoring hepatic function over time.

The advantages of piezoelectric gas sensors are high sensitivity, small size, fast response time, low power consumption, and robustness [31]. However, these piezoelectric sensors have a poor signal-to-noise ratio, as they operate at very high frequencies and require complex electronic circuits to delineate the signal response, making it difficult for them to act as a supportive element for an efficient e-nose system.

Therefore, the objective of this study was to determine, within the framework of an explorative study, whether liver dysfunction is generally recognizable and whether the level of this dysfunction can be classified utilizing a wearable semiconducting MOx gas sensor-based e-nose.

## 2. Materials and Methods

### 2.1. Electronic Nose and Signal Processing

In this study, a system called “LiverTracer” was developed. It is based on an e-nose system that detects changes in the VOCs from exhaled breath caused by liver dysfunctions and their severity. This system consists of a measuring head, which contains the sensor array, and a base unit for measurement control and data analysis (Figure 1).

The sensor head contains three active MOx semiconductor gas sensor modules (TripleSensor^®^, UST Umweltsensortechnik GmbH, Geschwenda, Germany). Each sensor module consists of three different gas-sensitive MOx layers that can detect reducible, easily oxidizable, and highly oxidizable gases. The selectivity and sensitivity of the sensor layers for different gas molecules depend mainly on the MOx semiconductor materials and specific catalyst additives used, and can additionally be varied by temperature changes. The latter are controlled by a platinum (PD) heater integrated into each sensor module. Depending on the type of gas, the gas molecules interact specifically with the surface of the different sensor layers, resulting in changes to their electrical conductivity. This conductivity (here measured as resistance) is registered and evaluated. According to the type of gas and sensor layer, concentration ranges from a few ppb up to the percentage range can be detected.

The used gas sensor elements (Triplesensor^®^) S1, S2, and S3 are realized through hybrid technology: they include a ceramic carrier substrate (aluminum oxide (Al2O3)) with a micro-structured PD thin-film layer, covered with a passivation layer, specific layers for contacts, as well as a gas-sensitive metal-oxide semiconductor layer (or layers) [32,33,34].

S1 is a ceramic MOx semiconductor gas sensor element with PD multi-electrodes, with a length × width × height (L × W × H) of 2.1 mm × 2.3 mm × 0.63 mm, respectively, with one sensitive MOx layer 2000C2+ (tin oxide (SnO2) thick film layer with a specific catalyst, for the detection of easily oxidizable gases, mainly carbon monoxide (CO), as well as hydrogen (H2) and ethanol (C2H5OH)). The processing of multi-electrode structure signals will be used for the detection of non-desorbing components/contaminations.

S2 is a ceramic MOx semiconductor gas sensor element UST Triplesensor^®^ (type 3A4P10), with an L × W × H of 2.1 mm × 2.3 mm × 0.63 mm, with three sensitive MOx layers: 2000C2+ (specific SnO2 compound with a specific catalyst and a thick film, for the detection of easily oxidizable gases, mainly CO, as well as H2 and C2H5OH), 3000C2+ (a specific SnO2 compound with a specific Pd catalyst and a thick film, for the detection of heavily oxidizable gases, mainly hydrocarbons (CxHy), and which is optimal especially for a number of carbon atoms (C1 to C8)), and 5000C2+ (a specific tungsten trioxide (WO3) compound with a thick film, for the detection of reducible gases, e.g., nitrogen dioxide (NO2)).

S3 is a ceramic MOx semiconductor gas sensor element UST Triplesensor^®^ (type 3A4P10), with a L × W × H of 2.1 mm × 2.4 mm × 0.63 mm, with three sensitive metal-oxide layers: 1000C2+ (a specific SnO2 compound with a catalyst and a thick film), 2000C2+ (a specific SnO2 compound with a specific catalyst and a thick film, for the detection of easily oxidizable gases, mainly CO, as well as H2 and C2H5OH), and 9000C2+ (a specific SnO2 compound with a catalyst and a thick-film, for the detection of long chain hydrocarbons).

The electronic microcontroller modules installed in the measuring head, with an analog-to-digital converter for each sensor element, control the heating temperature, the preprocessing of the sensor signals, the storage of the calibration data and the communication with the basic unit.

A spirometer “SPIROSTIK COMPLETE” (Geratherm Respiratory GmbH, Bad Kis-singen, Germany) was used as the basic unit. It contains a Windows 10 computer system. This device was modified according to our requirements. In particular, the sensor control, data storage, operator guidance (semi-automatic patient measurement), and data analysis were developed and integrated on the software side, as were the pump system for flushing and calibrating the measuring head on the hardware side. The principle of the measurement regime is shown in Figure 2.

After starting the system, it is checked whether a scheduled calibration of the e-nose is necessary to verify the correctness of the reference resistance values of the sensor layers to avoid measurement errors. For this purpose, a commercially available test gas (consisting of the components carbon monoxide, oxygen, and nitrogen) is used. Strong deviations of the measurement results from the resistance pattern typical for the applied calibration gas indicate the contamination or aging of the sensor layers. In this case, suitable countermeasures (cleaning, sensor replacement…) must be carried out.

If calibration is not required, or after successful calibration, preparation for the actual patient measurement begins. For this purpose, the operator selects an existing patient from the patient database (in case of repetition) or enters the required data for a new patient into the patient database and starts the patient data acquisition.

The processing of the patient measurement protocol (based on a predefined temperature control of the sensor heater optimized in preliminary studies [35]; see Figure 3a) is started with a cyclic thermal cleaning of the sensors until the sensor layers reach their original reference resistances (time-variable process). This is followed by the recording of the room air composition and the actual two patient measurements. By controlling the sensor heating temperature, it is possible to influence the sensitivity of the sensors for different VOCs (extension of the detection range). Burn-off cleaning phases serve to burn or evaporate impurities that may have adhered to the sensor surface. The measurement protocol has a duration of about 16 min. The temperature profile and the associated resistance data curves of all sensor layers are stored for subsequent analysis.

Data analyses were performed using MATLAB R2019a (The MathWorks, Inc., Natick, MA, USA). The 9 raw resistance waveforms were evaluated for outliers, technical problems, artifacts, and measurement errors. No measurement had to be discarded. For the analysis of the respective breathing air segments, the relevant 30 s segments were extracted from the measurement (Figure 3b, marked by vertical dashed lines). This was performed automatically based on the specified temperature measurement protocol, which clearly defines where the breath measurement starts and ends (Figure 3a, “bc1” and “bc2”).

The data analysis is based on the extraction of 10 features (time domain and nonlinear dynamics domain) from the resistance time series of the extracted breathing gas measurement cycles for each sensor layer.

In the time domain, the following features (Figure 4) were calculated:slope_startmax (Ω/s): slope from cycle start (Start) to absolute maximum (Max);s_slope_startmax (Ω/s): steepest slope of 1s duration from cycle start to Max;s_slope_startmax_pos (s): corresponding position of s_slope_startmax;s_slope_maxmin (Ω/s): steepest slope of 1s duration from Max to minimum (Min);area1 (Ω·s): area under the curve from cycle start to Max;area2 (Ω·s): area under the curve from cycle Max to midpoint;area3 (Ω·s): area under the curve from cycle midpoint to Min;area4 (Ω·s): area under the curve from cycle Min to cycle end;area3sec_9 (Ω·s): ninth subarea (24 s to 27 s); area under the curve is evaluated incrementally in 3 s subareas beginning from cycle start.

From the nonlinear dynamics domain, a feature of classical symbolic dynamics and one entropy measure were used. By employing symbolic dynamics [36,37], the original time series is transformed into a symbolic sequence and, thus, presented in a coarser form. Detailed information is lost, which allows the quantification of the dynamics contained in the time series. In the present study, for the quantification of symbolic dynamics of the resistance time series R of the breathing gas measuring cycles, the symbols 0 and 1 were assigned according to the following transformation rules:(1)$$0:\;R_{n+1}-R_{n}\leq 0,\;1:\;R_{n+1}-R_{n}>0.$$

Here, *R_n_* and *R_n+_*_1_ are the resistance values at the time points *n* and *n* + 1. While symbol 0 indicates decreasing resistance values, symbol 1 reflects increasing resistance values. Based on the transformed symbol string, words were formed consisting of two successive symbols. The frequency distribution of the word type 00 was determined (this was less dependent on minor fluctuations):*p*00—probability for the occurrence of the word type 00 within the resistance value time series.

The entropy measure, Renyi entropy, was calculated [37]. The density distribution (histogram) of resistance values in the resistance time series required for entropy calculations was determined using six classes. The optimal number of classes *k* was calculated using Sturges’ criterion [38]:(2)k=1+3.32∗log(N), N…number of resistance values.
Based on the density distributions, the individual class probabilities *p_i_* were calculated (with *i* = 1 to *k*), followed by the estimation of the following Renyi entropy measures:(3)$${\mathrm{Renyi}}_{-}\alpha \;\left[bit\right]=\frac{1}{1-\alpha }*\log_{2}\overset{k}{\underset{i=1}{\sum }}p_{i}{}^{\alpha }$$
Renyi entropy was estimated considering the coefficient value *α* = 4, which influences the weighting of the probabilities *p_i_* (weights larger fluctuations stronger than smaller ones).

### 2.2. Patients

A total of 30 participants were enrolled, 10 of them being healthy controls, 10 with compensated cirrhosis, and 10 with decompensated cirrhosis, between October 2019 and March 2020. Participating patients were randomly recruited consecutively according to availability in the normal care unit. Patients with ongoing acute-on-chronic liver failure, mechanical cholestasis, acute renal failure, malignant disease, severe cardiopulmonary disease (New York Heart Association classification severity level of heart failure NYHA III/IV (severe heart failure) [39] and/or chronic obstructive pulmonary disease (according to Global Initiative for Chronic Obstructive Lung Disease (GOLD) categories C (high risk/less symptoms) and D (high risk/more symptoms)) [40], and uncontrolled diabetes mellitus were excluded from the study. Control patients were either admitted to the hospital for elective hospitalization for non-liver disease (*n* = 8) or healthy medical staff (*n* = 2). Controls were matched for age, sex, and bodyweight. Decompensation was classified according to the Child–Pugh classification score (CPS) [41]. Patients that were classified as CPS B (significant functional compromise) or C (decompensated disease) were allocated to the decompensated group. In addition, patients with variceal hemorrhage were classified as decompensated.

The patients with compensated cirrhosis were male in 7 cases, had a median body weight of 94 kg and a median age of 57 years. Four of them were smokers. The etiology of cirrhosis was ethanol in 6 of these patients and four had other reasons for cirrhosis (2 viral hepatitis, 2 cholestatic liver disease). Patients with decompensated cirrhosis were male in 8 cases, had a median bodyweight of 80 kg and a median age of 62 years. Three of them were smokers and, again, the main etiology of cirrhosis was ethanol consumption in 8 of the patients (the others were 1 autoimmune hepatitis and 1 nonalcoholic steatohepatitis).

Control participants were male in 5 cases and had a median bodyweight of 81 kg. They had a median age of 58 years and one of them was a smoker. They had no history of known liver disease. None of the demographic parameters showed significant differences between the three groups. Vital parameters at inclusion between these groups did not differ as well (Table 1).

Relevant co-medication with known influence on intestinal flora and, therefore, on the results of the LiverTracer was analyzed. Lactulose was taken by 1 control patient, 3 patients with compensated cirrhosis, and 8 patients with decompensated cirrhosis (*p* = 0.009). Antibiotics were taken by 1 control patient, 3 patients with compensated cirrhosis, and 8 patients with decompensated cirrhosis (*p* = 0.016); however, the difference in antibiotics were caused by rifaxmin, which was taken by 1 patient with compensated and 6 with decompensated cirrhosis. Protone pump inhibitors (*p* = 0.262) and betablockers (*p* = 0.897) did not show differences between both groups (Table 1).

All procedures performed in the study involving human participants were approved by the Institutional Ethics Commission of the University Hospital Jena (5359-11/17), and were performed in accordance with the 1964 Helsinki declaration and its later amendments. Written informed consent was obtained from all individual participants prior to inclusion in the study.

### 2.3. Statistics

Statistical analyses were performed using IBM SPSS 21.0 (IBM Corp. Released 2012. IBM SPSS Statistics for Windows, version 21.0. Armonk, NY, USA: IBM Corp). Descriptive statistics were used to calculate means, standard deviations, medians, and interquartile ranges for all features calculated from the resistance time series separately for all nine sensor layers for respiratory gas measurement. The Kolmogorov–Smirnov test was applied to check the normal distribution of the features. The presence of statistical differences between the respiratory gas analysis characteristics of the control group (CON) and the two groups of patients with compensated (COMP) and decompensated (DECOMP) cirrhosis was tested with Welch’s t-test for normally distributed characteristics and with the nonparametric exact two-sided Mann–Whitney U test for non-normally distributed characteristics. A significance level of *p* < 0.05 was considered to be the criterion for statistical differences. Consistent with most of the published work on this topic, this paper presents only means and standard deviations for the identified features, regardless of the distribution or significance test applied, which improves the comparability of study results. Forward stepwise linear discriminant analyses combined with the leave-one-out cross-validation procedure were performed, and receiver operator characteristic (ROC) curves were calculated to assess the classification strength of the feature sets. Sensitivity (SENS), specificity (SPEC), area under the ROC curve (AUC), and accuracy (ACC) were determined for significant features and feature sets, each consisting of 2 or 3 uncorrelated (Pearson correlation coefficient) significant features. The resulting discriminant function analysis was then determined to be the classifier for automatic classification.

## 3. Results

We report below only the results of the first breathing gas cycle, as we did not find significant differences between the first and second breathing gas cycles. Let us first consider the classification results of the LiverTracer e-nose (Table 2 and Table 3). The separation of the patient groups (Table 2) from the controls was 100% successful in each case. Between the patient groups, a correct classification of 95% was achieved, where 90% of the patients from the DECOMP group and 100% of patients from the COMP group were correctly classified. Interestingly, these remarkable classification results were reached using only the features of sensors 1 and 3. Sensor 3 mainly contributed to the result. Sensor 2 did not make any significant contribution. Table 3 shows the descriptive statistics of those features that were automatically selected by the discriminant analysis to obtain the optimal separation results.

In Table 4, we included four clinical parameters for the stratification, which are based on the Child–Pugh score and represent different aspects of liver disease, including two laboratory values and two clinical aspects. Bilirubin, the end product of hemoglobin degradation, is cleared from circulation via hepatic elimination and, therefore, elevated in patients with cirrhosis and disturbed liver function. The international normalized ratio (INR), a marker of coagulation, includes proteins synthetized in the liver, which are therefore lowered in cirrhosis. Ascites is frequently present in advanced cirrhosis and is a consequence of cirrhosis-associated portal hypertension, while the occurrence of a hepatic encephalopathy is a typical complication of disturbed detoxification. We decided to skip the fifth parameter, albumin, as this also represents liver synthesis. Except for hepatic encephalopathy, no parameter was convincingly successful. While the controls could still be separated successfully, the detection of liver dysfunction severity was not convincing. The successful classification by hepatic encephalopathy is not surprising, since it was a component of clinical diagnostics.

## 4. Discussion

This exploratory pilot study extracted and analyzed unique VOC fingerprints in the breath of patients and provides initial evidence that breath VOC analysis using MOx sensors is a potential diagnostic tool for detecting liver dysfunction of different severities.

The sensitivity, specificity, and accuracy for distinguishing compensated and decompensated cirrhosis from healthy controls was 1.00 in all cases. Compensated and decompensated cirrhosis patients could be distinguished with a sensitivity of 0.90, a specificity of 1.00, and an accuracy of 0.95. Sensor 3 (with its three layers) showed the highest discriminatory power, and sensor 1, layer 1 could improve the result of sensor 3 by up to 5%. It was quite sufficient to evaluate only the first exhalation cycle of the patient. The inclusion of the second exhalation cycle did not bring any improvement.

In this study, we included patients with different stages of liver cirrhosis. Differentiation between patients with and without early stages of cirrhosis is challenging, but of great clinical importance. It is usually based on a combination of clinical, imaging, and laboratory parameters, but all of these can be influenced by non-liver related factors as well. Despite these weaknesses, the differentiation between cirrhosis and non-cirrhosis is of great clinical relevance, as the rate of postoperative complications and the mortality are higher in patients with cirrhosis [42]. However, the main predictor of these complications is the hepatic portal venous pressure gradient [43], which is not routinely measured. Using single laboratory parameters or clinical features does not result in the satisfying identification of patients with especially compensated cirrhosis in our study.

A study by Germanese et al. [44] that attempted to discriminate the severity of liver disease, particularly based on detected breath ammonia with MOx sensors, showed that the accuracy of discriminating between non-cirrhotic patients with chronic liver disease and cirrhotic ones was only 0.63, while that of discriminating between liver diseases and healthy controls was 0.81.

The generation of specific VOCs within the body can be the result of metabolic derangement, toxin or teratogen exposure, and finally microbiological processes [45]. Breath tests, which provide an indirect, non-invasive, and relatively low-input evaluation of various diseases, are used as diagnostic tools for quantifying the presence of one or more metabolites of a particular substrate in exhaled breath. Qin et al. [46] analyzed breath samples in hepatocellular carcinoma patients and controls by means of gas chromatography–mass spectrometry (GC/MS) combined with solid phase microextraction. Three potential VOCs, 3-hydroxy-2-butanone, styrene, and decane, were selected as promising biomarkers. A survey of other potential biomarkers in various liver diseases can be found in the publication by De Vincentis et al. [47]. Interestingly, alkanes (decanes) are precisely the group of markers that are particularly favorably detected by the sensors used in our study.

Even though these preliminary results are very promising, several limitations of this explorative pilot study are worth noting. First, it must be noted that the number of patients included is relatively small. However, it should be noted that this is a proof-of-concept study with a new sensor technology compared to previous studies [48]. It should also be noted that, in general, e-noses allow only indirect gas compound detection. In future studies, we intend to combine them with classical laboratory methods (e.g., GC/MS) to enable a direct assignment of biomarkers to the sensors. This would also have the advantage whereby the sensors could be optimally adapted to the pathology via the appropriate doping of the sensor layers. Another limitation of this study is that only a single measurement was performed per patient. Therefore, the system should be validated in a long-term and repeatability study. Additionally, we must mention that the influence of acute events, such as infections, was not studied in detail. This should be addressed in a subsequent study. Finally, MOx sensors also have drawbacks that are mainly related to the lack of sensor stability and the production of sensors with nearly identical sensor characteristics [13,49,50]. Among other factors, contamination and aging of the sensors may lead to short- and long-term drift of the sensors, causing differences in the measured sensor values compared to the originally measured values of new sensors, and reducing the accuracy of pattern recognition based on a trained pattern. Time-consuming recalibrations are often required to compensate for the drift [51]. Replacing a nearly identical sensor is usually difficult. Our approach to significantly reduce the drift and aging problems of the MOx sensors we use is to automatically assess the quality of the sensors before each breath measurement based on the resistance values of the individual sensor layers. In doing so, we compare these with stored threshold values of the original resistances (values of the sensor when it was newly installed). If there are deviations from the original resistance thresholds beyond a certain threshold, cyclic thermal cleaning is automatically performed until the sensor reaches the stored thresholds. If the thresholds are not reached within a specified cycle frequency, the sensor is recalibrated with a test gas. If all these measures fail, the sensor should be replaced with an adequate sensor with as close to identical resistance values as possible, and the e-nose may need to be recalibrated. However, throughout the study period, the values of our applied sensor array remained within the approved quality level. We therefore assume that the drift problem can be largely compensated for by sensor monitoring and calibration, but there remains some residual risk (especially in the case of sensor failure), the impact of which is currently being investigated in a validation study.

The results from this pilot study are very promising and suggest the principal suitability (especially by using the complex feature extraction method) of the MOx multisensory signals for the analysis of breath changes and, thus, for the identification of liver dysfunctions. Among the sensors used in e-noses for medical diagnostics, MOx semiconductor sensors are by far the most popular. They have high sensitivity, are durable, and, probably most importantly, are relatively inexpensive. Price is an important factor when considering large-scale commercial deployment, especially in developing countries. In addition, because they can operate in a wide range of relative humidity, they are particularly suitable for outdoor use [13,31].

In medicine and biology, e-noses are intelligent biosensor-based systems for the rapid detection, analysis, and classification of complex gaseous odors (usually as VOC mixtures of compound metabolite profiles). These instruments are innovative diagnostic tools with great potential for the non-invasive early detection of many types of diseases based on the analysis of VOC metabolites in the form of gaseous clinical samples [52]. They are inexpensive, have low operating and maintenance costs, and provide real-time analysis. Due to the growing demand for improved healthcare devices and procedures, the need for simpler and wearable e-nose systems that can provide fast and accurate diagnostic results and replace traditional, complex, often expensive and time-consuming clinical and laboratory methods has permanently increased. Such systems should non-invasively detect VOCs and accelerate on-site testing, allowing earlier diagnosis, faster treatment of disease, better prognosis, shorter hospital stays, faster recovery, and ultimately lower healthcare costs. Further development and point-of-care testing of new e-nose technologies and the development of standardized diagnostic methods will help bring these e-noses into routine clinical practice.

In summary, the multisensory analyses performed in this study based on a wearable MOx sensor array showed high separation accuracies of 95% to 100% between the studied groups. It was not only possible to distinguish liver dysfunctions of different severity from controls at 100%, but also to discriminate between the severities of liver dysfunction at 95% with a correct identification of 100% of all COMP cirrhosis and of 90% of all DECOMP cirrhosis).

Based on a semiconductor MOx sensor array, the wearable e-nose system for detecting disease—in this case liver dysfunction—offers significant advantages over conventional laboratory analysis and the use of other sensor systems when combined with the nonlinear processing of sensor signals. Our system thus represents a promising tool for distinguishing between patients with compensated and decompensated cirrhosis on a functional basis, and can thus make an important contribution, e.g., in preoperative workup or at the level of the general practitioner for the initial diagnosis and, thus, early detection of liver dysfunction.

## Figures and Tables

**Figure 1 biosensors-12-00070-f001:**
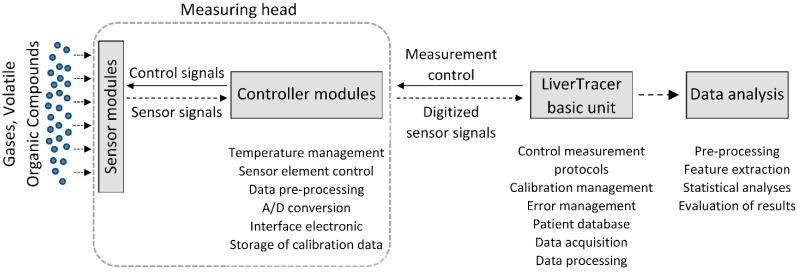
Setup of the electronic nose system “LiverTracer”.

**Figure 2 biosensors-12-00070-f002:**
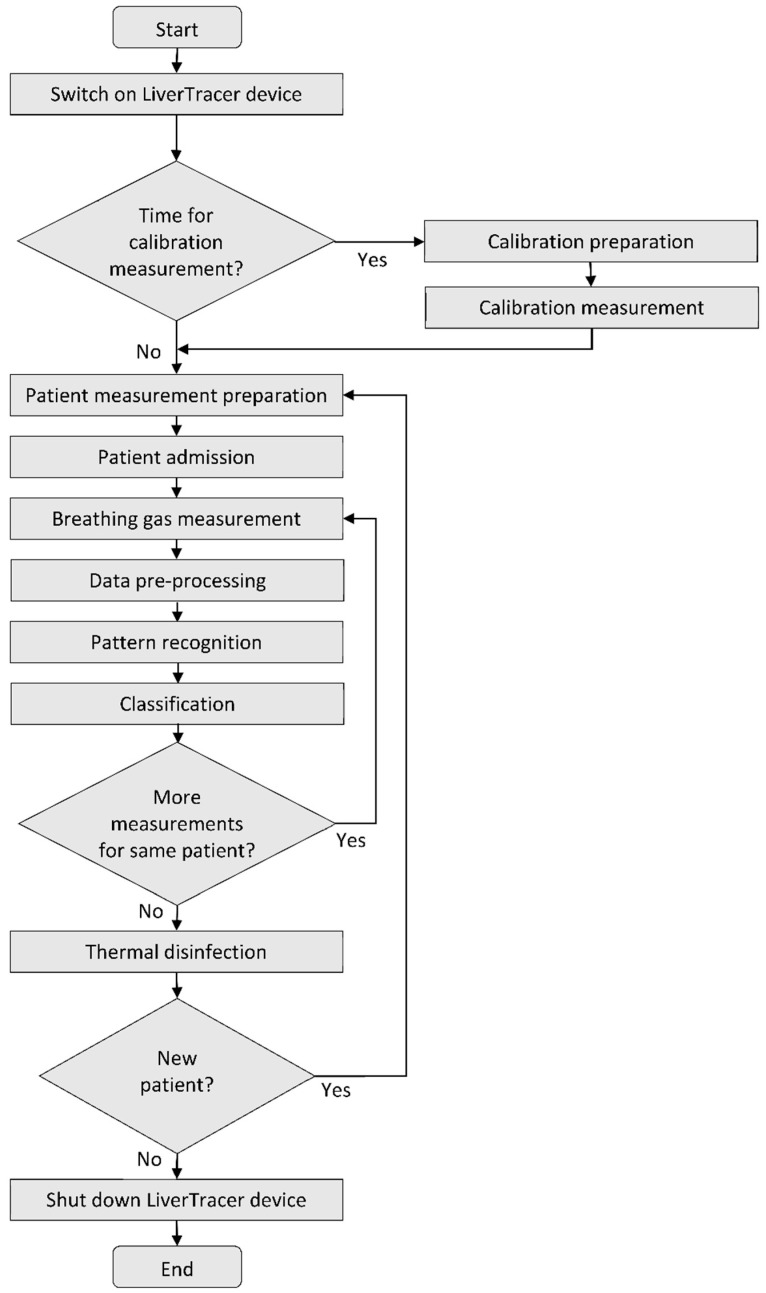
Flow chart of the LiverTracer measurement regime.

**Figure 3 biosensors-12-00070-f003:**
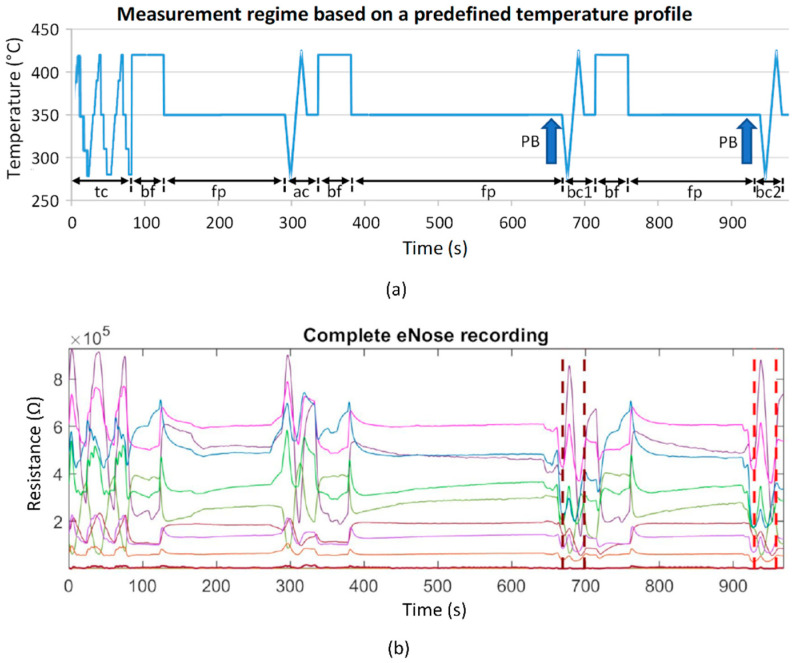
(**a**) Schematic representation of the measurement protocol based on a predefined temperature control of the sensor heater. It contains time-variable cyclic thermal cleaning cycles “tc”, burn-off cleaning phases “bf” (rectangle functions), subsequent flushing phases “fp” (horizontal lines), one ambient air measurement cycle “ac”, and two breathing gas measurement cycles “bc1” and “bc2”. The arrows mark the exhalation cycles (patient breathing: PB); (**b**) example of a recording of 9 sensor layer resistance curves. Vertical dashed lines mark the two breathing gas measurements.

**Figure 4 biosensors-12-00070-f004:**
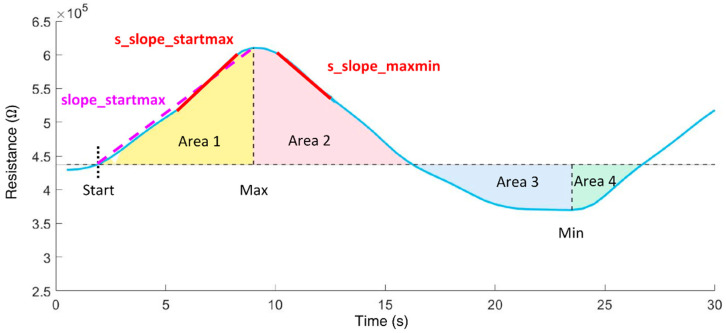
Time domain features extracted from the resistance curve of an exhalation cycle.

**Table 1 biosensors-12-00070-t001:** Patient data (values in parentheses represent the respective minimum and maximum values or describe percentages).

	Control (*n* = 10)	Compensated Cirrhosis (*n* = 10)	Decompensated Cirrhosis (*n* = 10)	*p*-Value
Sex (f/m)	5/5	3/7	2/8	0.500
Age (years)	58 (51; 65)	57 (52; 64)	62 (56; 67)	0.543
Bodyweight (kg)	81 (68; 96)	94 (79; 101)	80 (68; 97)	0.136
Height (cm)	175 (167; 178)	176 (167; 178)	176 (169; 181)	0.712
Smoker (n,%)	1 (10%)	4 (40%)	3 (30%)	0.450
Vital signs				
RR systolic (mmHg)	135 (118; 161)	126 (107; 155)	122 (103; 136)	0.266
RR diastolic (mmHg)	81 (76; 104)	76 (61; 92)	72 (63; 79)	0.146
Heart rate (pbm)	78 (67; 102)	85 (71; 88)	92 (81; 104)	0.212
Temperature (°C)	36.8 (3.4; 37.0)	36.6 (36.1; 37.0)	36.7 (36.4; 37.1)	0.523
Etiology of cirrhosis (n,%)				
Ethanol	N/A	6 (60%)	8 (80%)	0.628
Other	N/A	4 (40%)	2 (20%)	
Co-medication (n,%)				
Lactulose	1 (10%)	3 (30%)	8 (80%)	0.009
Proton pump inhibitors	5 (50%)	7 (70%)	9 (90%)	0.262
B-Blocker	5 (50%)	4 (40%)	5 (50%)	0.897
Antibiotics	1 (10%)	3 (30%)	7 (70%)	0.016
Rifaximin	0	1 (10%)	6 (60%)	
other	1 (10%)	2 (20%)	1 (10%)	

f—females; m—males; *n*—number of patients; *p*—significance.

**Table 2 biosensors-12-00070-t002:** Percentage classification rate of e-nose features. The optimal parameter set (consisting of either double or triple sets) is shown for each group comparison.

Group	Features	SENS	SPEC	ACC	AUC
CON—COMP	RS11_s_slope_maxmin (Ohm/s) RS32_area3sec_9 (Ohm·s) RS32_p00	1.00	1.00	1.00	1.00
CON—DECOMP	RS31_slope_startmax (Ohm/s) RS32_s_slope_startmax_pos (s) RS33_p00	1.00	1.00	1.00	1.00
COMP—DECOMP	RS32_Renyi4_entropy (bit) RS33_area2 (Ohm·s)	0.90	1.00	0.95	0.97

CON—control group; COMP—patients with compensated cirrhosis; DECOMP—patients with decompensated cirrhosis; RSxy—R denotes resistance measurement values of sensor layer y of sensor Sx (e.g., RS12 describes the resistance readings of sensor layer 2 of sensor S1); SENS—sensitivity; SPEC—specificity; ACC—Accuracy; AUC—area under the receiver operator characteristic curve.

**Table 3 biosensors-12-00070-t003:** Classification results of features automatically selected by discriminant analysis (mv—mean value, sd—standard deviation).

Group			CON	COMP	DECOMP
Test	Features	*p*	mv ± sd	mv ± sd	mv ± sd
CON vs. COMP	RS11_s_slope_maxmin (Ohm/s)	0.046	−86,258 ± 5225	−81,023 ± 5676	
	RS32_area3sec_9 (Ohm·s)	0.038	1,807,616 ± 207,540	2,071,884 ± 309,151	
	RS32_p00	0.017	0.336 ± 0.050	0.276 ± 0.045	
CON vs. DECOMP	RS31_slope_startmax (Ohm/s)	0.029	8901 ± 3207		6956 ± 1845
	RS32_s_slope_startmax_pos (s)	0.019	6.250 ± 1.161		6.900 ± 0.211
	RS33_p00	0.041	0.369 ± 0.045		0.319 ± 0.056
COMP vs. DECOMP	RS32_Renyi4_entropy (bit)	0.028		1.843 ± 0.386	2.179 ± 0.185
	RS33_area2 (Ohm·s)	0.131		48,252 ± 23,296	34,507 ± 14,547

CON—control group; COMP—patients with compensated cirrhosis; DECOMP—patients with decompensated cirrhosis; RSxy—R denotes the resistance measurement values of sensor layer y of sensor Sx (e.g., RS12 describes the resistance readings of sensor layer 2 of sensor S1); *p*—significance value; mv ± sd—mean value ± standard deviation.

**Table 4 biosensors-12-00070-t004:** Classification rate (in %) of the clinical parameters that achieved an overall accuracy for discriminating the groups greater than 50%.

	Categorized Bilirubin	Categorized INR	Ascites	Hepatic Encephalopathy
CON	100	86	100	100
COMP	10	40	70	100
DECOMP	90	60	50	50
ACC	63	59	73	83

CON—control group; COMP—patients with compensated cirrhosis; DECOMP—patients with decompensated cirrhosis; INR—international normalized ratio of blood clotting test; ACC—Accuracy.

## Data Availability

The data of this study are not publicly available due to the fact that the study has not yet been completed, and further evaluations are currently in progress. However, the data are available on request from the corresponding author.

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
