# Peer review of "Detection of Liver Dysfunction Using a Wearable Electronic Nose System Based on Semiconductor Metal Oxide Sensors"

_biosensors, 2022, doi:10.3390/bios12020070_

Round 1
Reviewer 1 Report
The authors presented very interesting studies of the application of an electronic device to the detection of liver disfunction by analysis of patients' breath.
The reported performance indicates that the proposed method allows perfect discrimination between cases of control, compensated cirrhosis, and decompensated cirrhosis. Such a result should be taken with caution. In Table 4 the authors present classification performance using commonly used clinical parameters and they don't allow such perfect classification. Patients with compensated cirrhosis are asymptomatic, non-invasive parameters all may be normal and liver biopsy would be required for diagnosis.
It is not clearly written but it seems that the results presented in Table 2 are based on the discrimination model performance evaluation is based on the same dataset on which model parameters are fitted. This means that probably there was no splitting of the dataset into training and validation/testing subsets. It would be more convincing if the reported results were based on an independent testing subset, or taking into account a small number of observations if the cross-validation procedure had been applied.
The results presented in the last row of table 2 seem strange.
The accuracy is 0.95 which taking into account the number of observations in the COMP+DECOMP groups seems that 1 of 20 patients was incorrectly classified. But in such a case the AUC should not reach a value of 1.00 as reported.
Also in this table, it is not clear to which group COMP or DECOMP is related sensitivity measure.
It is not precise in the paper but it seems that only one measurement of each patient was performed. I base this assumption on the numbers in Table 2, in which it seems that only one misclassification case was used. It should be explicitly written. Did the authors perform only one repetition of measurements for each patient?
The collection of data took several months from October 2019 until March 2020. Do the patients belonging to three considered categories were measured in random order? Since there were only 30 patients, with one measurement for each, it could happen that during such a long period the sensors drift is important order of measurement may be an important bias.
The authors use the discriminant analysis method, it seems that it is a Linear Discriminant Analysis method and should be explicitly written.
Reviewer 2 Report
Comments are attached below

Round 2
Reviewer 1 Report
The authors imporoved the manuscript and it can by published in present form.